# Learning to Adapt? Leave and Arrival as Major Psycho-Social Challenges for Newly Arrived Adolescent Immigrants in Germany

**Anke Wischmann** 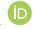

Institut für Erziehungswissenschaften, Europa-Universität Flensburg, 24943 Flensburg, Germany; anke.wischmann@uni-flensburg.de

**Abstract:** The aim of this paper is to analyse and discuss how learning is experienced by young, newly arrived immigrants in Germany. In particular, it addresses the connection between their experiences and the expectations of the German education system, as well as the connection between different kinds of learning experiences (formal and informal) in the context of adolescence. Adolescence is understood as an intergenerationally shaped psycho-social space of developmental opportunities. It is always affected and formed by aspects of social inequality such as milieu, gender, and race. Research shows that when migration takes place during adolescence, young people must cope with a "doubled transformation requirement" (King and Schwab). This paper discusses whether adolescent immigrants face even more transformation requirements—again, interwoven with learning—than two. Based on cases taken from two qualitative-interview-study samples, I reconstruct how coping strategies in terms of psycho-social development can be associated with different kinds of learning. Finally, I emphasise the responsibility of formal education as a supportive framework and stabilising factor for child and adolescent immigrants after they reach Germany.

**Keywords:** adolescence; newly arrived immigrants; learning; Germany; adaption

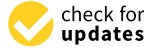



## 1. Introduction

In 2015 and 2016, about 1.2 million people came to Germany as refugees [1,2]. A total of 36.7% of the 2016 asylum applications were submitted by people between the ages of 11 and 25, including accompanied and unaccompanied minors from various countries, most of whom came from Syria and Afghanistan (ibid., p. 21). After a phase of mostly constructive debate, public discourse has shifted into a more problematic, if not downright pejorative mode. The social frames [3] this discourse creates for young immigrants in Germany thus tend to be rather deficit-oriented or even discriminatory [4].

After peaking in August 2016, the number of immigrants dropped and remained constant until February 2022 [5]. This was mainly due to measures limiting further immigration rather than to any lessening of their causes [6–9], as the actual situation shows. In 2022, about 1 million people from Ukraine came to Germany. It can be assumed that in the future, a large number of people will continue to come to Europe for different reasons and that, as a result, young immigrants will have to be integrated into the German education system.

The German education system is highly selective, since pupils are separated into different tracks after grade 4 [10,11]. The specific configuration differs between the 16 federal states, yet, in most states, two to three different tracks are present, only one of which leads to further academic education. For newly arrived immigrants, so-called Welcome or Preparation Classes were established in 2015, mainly to provide German-language skills [12]. These classes are mainly situated at primary and lower-track schools [13]. Recent figures suggest that young immigrants are underrepresented in academic tracks and higher education (ibid.). Hence, the goal to integrate immigrants in terms of equal achievement and educational participation was missed.

This situation can only be tackled successfully if the efforts that must be undertaken in order to achieve integration are determined, not only in terms of the immigrants' formal education but also in terms of their informal educational experiences. For adolescents who leave their countries, the context is one in which formal education has been interrupted or even stopped altogether [14–19]. Therefore, researchers need to consider the learning experiences of adolescent immigrants from a broader perspective, beyond that of formal education, in the context of the social conditions to which they are exposed. This perspective follows current educational research on the importance of recognising both informal and formal learning among adolescent immigrants [20].

The UNHCR differentiates refugees from migrants by stating that those in the former group cannot return to their country of origin and, hence, must be protected [21]. In practice, however, which of these two statuses an immigrant receives depends on their country of arrival. In many cases, the question of who actually gains legal status of "refugee" is quite controversial. In this sense, the differentiation between refugees and migrants is fluid rather than fixed. Furthermore, the statuses of different migrant groups differ greatly, such as between European and non-European immigrants, but also between other countries of origin, such as between Syria and Ukraine. The latter are not required to apply for asylum, whilst the former are, and the latter have the immediate right to work in Germany. Due to these discursive formations, I will not refer to only one status, but to the notion of newly arrived adolescent immigrants.

Young immigrants are hence socialised in specific—and rather heterogeneous—ways. This socialisation is related to particular kinds of learning experience during what scholars recognise as a sensitive phase of personal development. Research shows that when migration takes place during adolescence, the young people involved must cope with a "doubled transformation requirement" [22]: they must "transform themselves" into adults while simultaneously experiencing and adapting to the transformation of their environment and living conditions.

To reconstruct how learning is experienced by young immigrants and, at the same time, how it is interwoven with socialisation, a secondary analysis of the samples of two qualitative interview studies was carried out. Both studies had an open qualitative design and included adolescents between 11 and 17 years of age. The first study focused on young adolescents with non-specific backgrounds concerning migration to reconstruct the interrelations of formal and informal learning [20,23], and the second study focused explicitly on newly arrived adolescents [24,25]. The total number of interviews from both studies was 26; 15 interviewees were immigrants of first and second generations, and seven of these 15 interviewees immigrated with their families. The interviews were analysed by using a reconstructive method [26,27]. Hence, all the interviews included learning experiences, whereas most of the interviewees were positioned as migrants due to their families' migration, and seven of them had personally experienced migration. In this paper, I will focus on the typical features of learning when migration and adolescence coincide.

The research questions that are discussed here concern, first, how newly arrived (accompanied) adolescent immigrants experience learning while facing migration and adolescence, and second, how this learning is interrelated with the (prescribed) status of being a migrant. Hence, the discussion in this paper only includes immigrant youths, not second-generation immigrants.

## 2. Theoretical Framework: Adolescent Learning in the Context of Leave and Arrival

According to King [28], adolescence is an intergenerationally shaped psycho-social space of developmental opportunities. Adolescence is always affected and formed by aspects of social inequality, such as milieu, gender, and race [29,30]. Therefore, the constitution of adolescence differs, depending on how the young person is positioned socially at the nexus of these different variables. If the social position of an adolescent is precarious and they experience discrimination, the quality of the psycho-social space for such developmental opportunities is impaired.

Particular challenges for psycho-social development emerge when migration takes place during adolescence [31–33], and these challenges become even more acute when the migration represents a flight or forced migration from the migrant's homeland. The young person experiences two levels of separation, which cause specific psychological and social changes [22,34]. In addition, the legal status resulting from immigration throughout adolescence can become a master status and, hence, a major challenge [35,36], when it is not clear, insecure, or even illegal. However, for the German context, there are relatively few empirical studies of these changes [37,38], particularly against the current socio-political background, which involves the large number of young refugees who have arrived in Germany since 2015. Studies for the international context are more abundant (see [39] (Canada); [40] (Australia); [41] (USA); and [42] (Greece)).

A number of discourses, such as those centered on educational and political matters, refer to adolescent immigrants [43]. Their focus is on the integration of these young people. Following Ager and Strang [44], successful integration involves ensuring that young migrants receive appropriate access to work, housing, education, and healthcare. Furthermore, the legal status of migrants in their country of arrival, the establishment of social relations in that community, and the dismantling of so-called structural barriers, such as language learning, are additional relevant factors. Integration is seen as a challenge for both young people themselves (see, for example, [35]) and their society of arrival [45,46]. Media and political references to these young people most often note their inadequate German-language skills, their often low (or at least insufficient) level of education [47], and alleged cultural differences [48]. By contrast, structural problems present in the society of arrival, such as the "monolingual habitus" [49] of its educational system, or the intentional and structural discrimination against refugees within the educational system, the labour market [50], public administration [51,52], and the housing market [53], are less frequently addressed [54,55]. Nevertheless, these young people often report experiencing various kinds of discrimination, as well as the restrictions that their legal status imposes on them regarding their educational path and future planning [18,35,56]. This is particularly apparent for illegal or unauthorised young immigrants, who are in the position of not being able to plan an educational path or future in any way.

In addition, social–psychological research points to difficulties in dealing with trauma and multiple loss experiences [57–60]. These experiences become important for child and adolescent immigrants once they begin participating in the educational systems of their societies of arrival. To avoid retraumatising these children and adolescents, and to support their development of coping strategies, it thus seems important to expand our knowledge of their experiences [37,61]. At the same time, studies show that refugees are highly ambitious with regard to their educational and work goals [47].

It should also be recognised that newly arrived immigrants in general, and newly arrived adolescent immigrants in particular, are a very heterogeneous group (ibid.). Not only are there large linguistic, ethnic, and cultural differences between those in this group, but also many different causes of flight, escape routes taken, and—not least—different conditions in their countries of origin [62]. Even when examining the experiences of immigrants of the same age, large disparities emerge and must be considered. Learning experiences in adolescence are fundamentally linked to the specific psycho-social conditions to which individual adolescents are subjected [20,23].

Learning is understood here as a complex world- and body-bound process, in which knowledge and skills not previously available are developed (ibid.). This process includes small-scale learning processes in which concrete knowledge or abilities are gained, as well as those that result in far-reaching, long-term changes in the world and in one's self-awareness [63,64]. However, it is doubtful that learning processes always take place in the same way on an inter- and intra-individual basis [65,66]. Learning is to be understood as a subjective experience that is both body- and subject-related and is always to be seen in its relationality and contextuality. Thus, it has to be reconstructed on a case-by-case basis.

If learning is context-bound and understood intersubjectively, it is necessarily interwoven with and conditioned by socialisation. This is true for implicit learning as well as for the formal and informal learning that take place on a conscious level. While formal and informal learning differ in some fundamental ways, they are nevertheless interrelated [23]. Since learning is designed differently between individuals, it must also be assumed that it takes place on an age- or life-phase-specific basis. Learning in adolescence develops in specific ways, as does the relationship between formal and informal learning.

Adolescence has to be considered as a normative, discursively generated concept. Gradual individuation during this life phase, as the individual separates from their parents and from their own childhood, is a normative expectation connected to Western concepts of individualisation [67]. This refers not only to the expectations of the adult generation, including the researchers who study the subject, but also about those of the young people themselves. Nevertheless, these expectations can be experienced as particularly challenging in situations of social disadvantage and discrimination [68] and in contexts in which adolescence and migration merge [32].

Some empirical studies point out that informal learning is of great relevance to both the personality development and formal educational achievement of young people (e.g., [69–71]). It has also been shown that not every form of informal learning is equally helpful in this regard; this depends to a large extent on the social background of each adolescent. This is particularly evident in studies focusing on the educational significance of the family and peers during adolescence [72–74]. Essentially, the question is not whether informal learning takes place, but what is learned and how, and whether this particular learning is recognised as relevant to formal education or, more concretely, to the adolescent's educational achievement. While not surprising, these results need to be considered by researchers, especially since studies show that extra-familial and extracurricular social spaces are gaining importance as socialising entities in adolescence [75]. However, there is still a dearth of studies on the specific relationships between informal and formal learning, in which social conditions are taken into account, and research linking such studies with the adolescent migratory experience in general, and experience of flight and forced migration in particular, is still pending.

Studies on the relationship between immigration status and formal education clearly show that the former has a significant and mostly negative impact on the latter (e.g., [76]). Even young people who are not themselves immigrants, but rather belong to the second or third generation, perform worse in the educational system than those without a migration history (ibid.). If they have migration experiences of their own, these experiences have a massive impact on their educational trajectories, which also includes non-formal and informal education [77]. When the resident status is unclear or illegal, the situation worsens, because—as Gonzales pointed out—immigration becomes a major status that affects all parts of life and overshadows all kinds of activities [35,36]. Hence, the psycho-social costs rise and the effort to cope with everyday challenges increases.

King and Schwab [22] assume that in the context of flight in adolescence, a doubled transformational experience sets in, one that is neither negative nor positive to start with. Rather, it starts from complex requirements, which can be processed in very different ways. Whether any given adolescent succeeds in carrying out these transformations depends on aspects of their specific situation (such as their family situation), the itinerary of their flight, and their experiences in their society of arrival [30,57]. In addition, Geisen [78] notes critically that adolescents with migration experiences are on potentially unsafe ground because they must position themselves and behave in ways that are both migration-related and adolescent-specific. In particular, when adolescence is shortened or restricted by flight experiences, this usually has negative consequences for the adolescent's development and education (ibid., p. 40).

### 3. Methods and Materials

Neither of the two studies on which this paper builds were originally focusing on the learning experiences of newly arrived adolescents. The first aimed to explore the interrelations of formal and informal learning in early adolescence [20], and the second was a pilot study on adolescent immigrants and their experiences in the context of migration and arrival in Germany [24,67,79]. Whereas the first study included immigrants and non-immigrants, the second focused on newly arrived young people. The national backgrounds of the adolescents were diverse: three came from Eastern Europe, three from the Middle East, and one from Africa. Both studies used the same methodical design, namely narrative interviews [80], which encouraged the adolescents to tell their stories.

All interviewees were enrolled in compulsory education and aged between 13 and 16 years. The contact was initialised via teachers, social workers, or educators who enabled information events in informal school-related settings, where I was able to explain my work and ask for participants. The interviewees later on contacted me to comply and take part. Time and space were determined by the participants, as well as whether they wanted company of a friend or family member. In the first study, I asked all students to participate and, in the first study, only those who were newly arrived in Germany were included. All were fluent in German. Those who were not might have decided not to participate, even though translation was offered.

Even though the initial research questions of the studies were different and specifically aimed at learning, the data are appropriate to answer the research questions concerning learning experiences in the context of leave and arrival discussed here. Both studies featured open-interview designs, meaning open questions that invited the interviewees to speak about themselves and their stories. Moreover, the interviews were analysed reconstructively and, hence, word by word, closely to the text, so the inherent meaning was extracted and only in the next step contextualised with theory. Thus, the fact that similar logics of experiences emerged across the studies underlines the objectivity of the study and reduces researcher bias (at least to a certain extent).

As interviews had no set time limit, their durations varied, lasting from 30 to 60 min. The parents of interviewees, who were informed of the purpose and format of the interviews in different languages, consented to allow their children to take part in the study. As all participants had been in Germany for a year at least at the time of the interviews, they had already attended a so-called preparation/welcome class. The interviews were analysed using a reconstructive method that focused on both the content of the narrations and the form/structure of what was said, as well as how these two were related. This perspective enabled me to gain access not only to manifest meanings, but also to latent or implicit meanings.

The interviews were analysed by using the case-reconstructive method developed by Rosenthal [27]. A distinction was made between an experienced life story and a narrated life story, so that content and case structure could be contrasted. In this way, specific features and references to implicit aspects could evolve. The purpose of the reconstructive method is to gradually reduce the number of possible readings of a text in the sequential analysis, and to establish the specific selectivity of the speech [81]. In this context, the linguistic features of the interviews (grammatical abnormalities, code switching, etc.) are considered [82].

Rosenthal [26] focuses on the relationship between the past and the present of the narration. To this end, she uses the differentiation between a narrated and an experienced life story and assumes a dialectical relationship between experiencing, remembering, and narrating. Thus, she does not assume that narrative reflects the objective past, nor that only the present perspective determines the representation of the past. Her perspective encompasses not only lived experiences that are always located within a prefigured framework, but also social conditions and structures that affect entire generations.

The interview is taken as an overall construct. It is assumed that the narrating subject represents their own development from the perspective of the present, and that this representation is irreducible and cannot be torn from the context of meaning.

The procedure of the analyses includes, first, the analysis of biographical data (dates, places, events) without the consideration of the subjective interpretations of the narrator. Second, the interview text is divided into sequences, according to the text structure and content. The aim of the analysis is to determine which mechanisms control the selection and design, as well as the temporal and thematic linking, of the text segments. The third step is the reconstruction of the case storyline according to the biographical data and the subjective manner in which the story is told. Fourth, particularly important interview sequences are subjected to a micro-analysis (meaning, word by word). Fifth, the autobiography and the narration are contrasted (what is said, in which way, and why). Subsequently, a case-structure hypothesis can be formulated. Finally, the analyses of several cases are condensed into a typology.

Keeping in mind that the narrated story describes the adolescent subject's biographical self-presentation in the present, my analysis sought to understand how the narrators presented themselves. Even if the narrative contradicted the biographical data of the story experienced, my goal was not to evaluate a 'true' story in relation to the more or less fictitious presentation, but rather to understand why the story was told a particular way and not in another way. Furthermore, central text passages were subjected to a detailed analysis in the sense of objective hermeneutics [81]. This approach does not seek to generate certain categories of adolescent learning in the context of migration and arrival. Rather, by reconstructing the case structure, it aims to work out specific experiences, as well as the relationship between different forms of learning, experiences, and spaces. When examining the overall sample of both studies considered here, it was stated that there are special features of adolescent learning [20]. By contrasting cases of young immigrants with those who grew up in Germany, it was found that there are particular features concerning learning that they have in common, even though the stories and experiences of migration differed greatly. As stated above, the status of immigrant has a major impact on an individual's everyday life; however, we need to differentiate between an assigned status (by law, teachers, or administration) and the self-positioning of the immigrant. It was found that, even when the immigration was voluntary for the family or the parents, it was in all cases experienced as forced, since they were not involved in the decisions and felt powerless.

Clearly, due to the small number of cases, the results need further exploration. Therefore, a study is planned with an explicit focus on migrant education. Another challenge resulting from the type of reconstructive research I applied is language. This is because, first, in the context of the interview, only students who speak German can participate, or translation is needed. Second, the interviews must be translated into English for publication. Therefore, translation is an additional course of interpretation. In various multilingual interpretation settings, the interpretations were therefore intersubjectively validated. Nevertheless, errors are possible and can only be reduced, not eliminated.

## 4. Results

In the following section, the results of the secondary analysis with the focus on newly arrived adolescent immigrants are presented. The typical features of adolescent learning in the context of immigration and arrival that were reconstructed in the interviews show a common structure and include similar aspects; hence, significance can be assumed. To illustrate and substantiate the findings, anchor sequences from the interviews are cited.

The cases that are considered are anonymised and called Vladimir, Zola, Peter, Yusuf, Elisabeth, Janis, and Raul. All of them came to Germany while approaching adolescence, with their families, or parts of their families. The legal status of the young people concerning their residency differed, as did the conditions of their immigration. Hence, the factors that they had in common were that they were all relatively new to Germany (1 to 3 years), they all had to learn German after their arrival, and they all attended a so-called Preparation

or Welcome Class. The immigration took place during adolescence or late childhood. The cases are not presented individually; instead, I present the results of the analyses of these cases concerning the typical features of learning in the context of migration and learning.

There are three overall features in the presented sample that can be used as a frame to examine the significant aspects of learning for this particular group and, hence, to establish their needs in terms of education.

1.  Learning to be a migrant;
2.  Becoming normal;
3.  Learning to integrate and to achieve.

The first feature refers to the fact that these young people who fled or emigrated with their families were not involved in the decision to leave their home countries. Regardless of the objectives or reasons of the families (war, poverty, or persecution—or even unknown or implicit in the interview), all of the interviewed adolescents said that they had to subject to the decisions to leave. Zola, for example, when asked for the reasons for their emigration, stated the following:

> "Actually, my mother did not tell me why we come here. [I: mhm] But then one day she said to me uhm that we are flying to Germany [I: mhm], so she actually wants to visit other countries [I: mhm] . . . And that's why we came here." (Zola, 15)

The events that took place and the actual reasons appear to have been blurred, and it is implied that the mother simply wanted to travel as a tourist, which is not plausible in the context of Zola's coming to Germany as a refugee and living in a shelter for months, waiting for acceptance and for proper housing.

Elisabeth told an entirely different story. She was left behind by her parents with her grandparents for several years, so that her parents were able to work in Germany and make a living for the family. This would be legally defined as labour migration:

> "@ . . . ehh . . . so@ my parents have lived here for a really long time, about fifteen years. And I came to Germany two years ago because I lived in Romania . . . with my grandma and grandpa . . . like my sister . . . and my parents wanted us to live together like a normal family because it was really hard to live without my parents. So that was about six years, I don't know. (takes a deep breath) And then . . . I've been living in Germany for two years. That's how I came to Germany." (Elisabeth, 13)

Elisabeth was left behind, and then taken to Germany. As she makes very clear that living with her parents was very important to her, she had no choice but to follow them and become a migrant. Hence, Elisabeth's migration, which she did not choose, could have meant leaving together, following her parents after some time, or migrating first and waiting for her relatives to follow later on. This case refers to the research field on transnational childhoods [82]. This research discusses different experiences and strategies of transnational families in light of global migration and economy and shows that the decisions made by families in these contexts are complex. However, it is usually the parents who decide whether to migrate or to stay when migration for children takes place.

In Vladimir's case, the single mother left first, leaving their two sons behind with their grandmother, and the reasons for this are not clear:

> "Yes oh so in the beginning it was so that my mother came to Germany [I: mhm] so she was alone here for a year. [I: mhm] Then I was alone with my little brother and with my grandmother [I: mhm] at our house. I came from Ukraine [before the war] [I: mhm]. And yes, we waited a year until we can drive to her . . . um. [I: mhm] And then we went to her and yes. Actually, this was all new to me and so I've never seen something like that. [I: mhm]" (Vladimir, 16)

Hence, becoming a migrant was in all cases related to a feeling of powerlessness and the inability to act. This is, of course, solely from the perspective of the child, not the

mother. However, research on transnational motherhood powerfully shows the difficulties experienced by mothers when they feel obliged to leave their children behind [83]. Hence, mothers or parents might feel powerless in their situation but, structurally, they have a choice and decide, while their children do not. Therefore, regarding children and adolescence, an inter-generational difference needs to be considered to understand the perspectives of young immigrants, not least when they enter school shortly after their arrival. In terms of learning, this means learning to be a migrant or, as Mecheril [84] puts it, "the Other of Migration." This means to recognise a severe shift in social position and the experience of not-belonging paired with a lack of language skills, which is accompanied by feelings of social isolation.

> "Um, so I didn't talk at all, although I already know a bit of German, uh, so I could, . . . and I don't know, later I started to—talk more, . . . uh . . . to get in touch more, uh . . . yes I think that was because I wanted to talk to people because, . . . I- it wasn't so nice to be alone all the time. [I: mhm] and yes. I learned German to be able to communicate with my friends." (Elisabeth, 13)

Hence, learning German as is required by institutions and the host society—is understood as a major task. The inability to speak properly or being addressed as someone who cannot speak properly can negatively affect self-perception:

> "But yes, I just had the feeling that I wouldn't understand anything. ( . . . )Yes, and I have no friends, I was at home all the time. And. there was I was really sad. [I: mhm] Yes I wanted to go outside (?) fresh air and I couldn't make any friends. And because of that I had also lost 22 kilos in two months. ( . . . )" (Janis, 16)

Secondly, the adolescents had to restructure their position within the family and the adolescent space for opportunities. After arriving in Germany, they worked on the reinvention and stabilisation of their everyday life and strove for normalisation.

> "And . . . yes, then I went to school, at the beginning of the beginning it was hard for me, because I cannot say a word, nothing. [I. mhm, mhm] And . . . yes got to know my friends there. And yes. So, I was in the beginning in a—what's that again- prep class. And now I'm here for this school. For two years. [I: mhm, mhm.]" (Vladimir, 16)

As with the others, Vladimir experienced his arrival in Germany as strange, particularly concerning the foreign language that he was forced to wrestle with daily. The isolation the adolescents felt because of their inability to communicate drove them to learn German—in many cases, such as those of Vladimir and Elisabeth, very quickly. The international preparation class offered a context for the participants to socialise, make friends, and improve their German. Here, different learning experiences can be reconstructed. Learning relates to re-integration, and several kinds of learning experience are described in this context: (1) language learning, in both formal (preparation classes) and informal (peer interactions) contexts and (2) learning to adapt to new social and cultural expectations, particularly concerning school and the prerequisites for educational achievement. The latter is very important as a basis for future economic security. In keeping with this priority, excelling in their studies appeared necessary to stabilise Vladimir's psycho-social situation, including his familial situation. Vladimir explicitly tended to normalise his situation, noting that it was not different from that of other families living in Germany. Hence, learning (3) means to normalise and to adapt.

Thus, Vladimir focused on normalisation and educational achievement, since each of these was necessary for the success of the other and, ultimately, to stabilise and normalise his living conditions. "Normal" means the elimination of any differences between the adolescent and his/her family and their society of arrival. In order to stabilise the family situation, the adolescents became the "bearers of hope" for their families. Whereas the parents were in many cases degraded in terms of their professional recognition and job, the children were, due to their obligation to attend school, immediately involved in German

institutions and, in all cases, acquired the German language much more quickly than the adults. Thus, the power relations within the families transformed, as in Peter's case:

> "Well, my mother, she was an accountant and manager in Russia. [mhm] And now she's learning German and then she wants to become a manager or an accountant. Um, my father, he's in Russia? He works in a car company, I don't know exactly, something with cars, I think. Uh, yes. And my stepfather, he, ehm, (.) eh, he separated from my mother and now there are, ehm, three of us here. Me and my little brother and my mother. My brother, his name is Manuel and [mhm] he has turned four? [mhm] and @ja@ goes to kindergarten and is also learning German there. [mhm]" (Peter, 14)

It was no longer the parents who take care and provide resources. Rather, they all became equal as students with aims, concerning first language skills and then professional aspirations. The former position of the mother as an accountant in Russia no longer counted anymore. As a result, the educational achievement of her children became very important. This leads to the next feature, described below.

Thirdly, to successfully meet the described requirements, the adolescent immigrants needed to develop a strategy of learning that involved formal and informal learning in a fruitful balance, with the emphasis on formal learning, as a key to educational achievement in the host society. The strategies differed greatly between the cases. Some explicitly focused on school and worked hard to obtain a degree. Others used social learning to cope. For them, learning needed to be framed by relationships to significant others, and individual preferences were secondary to them. Hence, learning is embedded in social relations. This means that for adolescent migrants. formal learning structured by teacher–learner relations deeply depends on teachers and peers.

Thus, learning in these cases can be reconstructed as (1) learning to cope with unexpected changes in life, which are connected with separation, loss or the threat of loss. Learning the language (2) is important, first and foremost, because it enables the migrant to socialise. In Zola's case, it appears that the various languages spoken by her and her mother are not recognised as useful in the German context. They speak French, English, Yoruba, Arabic, and other African languages that are not specified in the interview (3).

> "Yes, [I: yes] I speak French, my mother tongue is Yoruba [I: yes] and they had another one but that—I can understand him when others speak, but I can't yet— speak like that. [I: mhm, mhm.] And I know a little Arabic too. I'm a Muslim and we also read the Koran, so I can speak a little Arabic. And many of the languages of Benin. [I: mhm] I can still. [And German. [I: mhm] And a little English. [I: mhm] [laughs]" (Zola, 15)

The three features that were reconstructed all helped the adolescent migrants to adapt to a new situation that affected their entire life and social situation, as well as their psycho-social space. Therefore, the overall requirement that was recognised by all the interviewees considered here was to learn to adapt. Even though the strategies of adaptation differed greatly between the cases, the requirements anticipated by the adolescents must be considered by educational institutions to meet their needs and to help them gain stability and the ability to act.

## 5. Discussion

Even though the cases were very different in terms of objective dates and their subjective positionings towards migration and learning, they also presented similarities. I interpret the latter as significant for adolescent immigrants who are forced to emigrate from what is, for them, not an obviously threatening living context under what are, for them, not transparent conditions. I discuss this by referring to the "doubled transformative challenge" [22] concerning the psycho-social space of opportunities and learning as a relational experience.

All the adolescents considered here were ripped out of their familiar lives and separated from their extended families by adults, who neither informed them in advance about their plans nor ever offered an explanation. This must have involved the loss of ground and perspective at the same time. This I interpret to be a major psycho-social challenge for the adolescents, because this experience is connected to a feeling of powerlessness which is counterproductive to the development of individual agency, which is crucial for adolescent's identity formation. If the transformative demand of adolescence doubles in any situation in which adolescence and migration coincide, in that the transformation of one's own childish identity into a future adult identity and the transformation of one's living context (with respect to language, culture, and often the socio-economic status of the family) occur at the same time, then in the presented cases we have been discussing, the transformative challenge is—at least—tripled. Hence, the challenges on the path of individuation and—again—agency are multiplied, while the (psycho-social and material) resources are limited. This refers to the third dimension, the transformation of the psycho-social space of opportunities, since it is an intergenerationally shaped space. King [28] describes an "inter-generatively framed individuation process" that takes place within this space. It is designed in a specific way, in accordance with the milieu, gender, and culture of each individual, and the quality of their intergenerational relationships—which are, themselves, at the same time, transformed within and by it. Stability during this profoundly unstable life phase is ensured here, above all, by members of the adult generation. Hence, the adults involved need to grant adolescents space for opportunities. However, if the family and other adults and institutions in charge (i.e., schools) do not provide this space, precarious self-positionings are already threatened, because the opportunities for grounding and orientation are fragile, if not non-existent.

Hence, the stability needed to develop agency and, with it, learning aspirations is not given. The family situations in the scope of migration are turned upside-down. At the same time, the adolescents must give something to their families: absolute loyalty and faith—a leap of faith, in fact, because they are unclear about the reasons for their own migration and their parents' motives. In this situation, it is evident that these displaced adolescents need to stabilise their relationships and, therefore, their space of opportunities. Thus, the adolescents are brought into the position of stabilizer, and the intergenerational relationship is reversed.

## 6. Conclusions

Adolescent learning as a relational experience is affected, if not shaped, by experiences of migration and throughout adolescence. Obviously, for all newly arrived adolescents, school plays an important role in their experience and the response to their arrival in Germany, as well as their ability to integrate into this new culture. Even though informal learning settings are also involved, the school has a crucial function as a scaffold, especially given the many losses these young people must endure and the unspoken migration-related conflicts within their families. Thus, school has a fundamental function as a stabiliser for the adolescent space of options, and adolescent learning in particular, for those who have recently arrived in Germany. This is likely to also be true for other countries of arrival.

Hence, teaching German and improving scholastic performance are not the only—and maybe not even the most important—tasks for school educators who work with newly arrived adolescent immigrants. It is also vital that these teenagers are provided with stable social frameworks and stable relationships so that they can resettle after coping with the demands of a double transformation and successfully complete the necessary socialisation and self-formation processes. Therefore, the concept of adolescence itself needs to be reflected on in the context of learning and immigration.

If the adolescents in this study were to receive such support—that is, stable and reliable learning settings—they would eventually profit from it. Although language learning is critical to their survival and success, it is nevertheless always connected to the provision of a supportive and stable educational setting. Given the triple transformation that these

adolescents already face when they arrive in Germany (self-transformation, adaptation to a new homeland, and adjustment to a new family situation), further transformation challenges in and presented by the school system into which they are placed must be avoided. As Bergnehr ([57], p. 2) states, "[f]or immigrant families undergoing resettlement, change and adaptation are particularly pertinent." To avoid further experiences of separation and loss for this particularly vulnerable group of students, the school system and the adults who are involved in it and responsible for ensuring that adolescents receive the requisite "spaces of opportunities" must become aware of their role in fostering the social integration of newly arrived adolescent immigrants. Considering the number of immigrant students who are entering the German educational system and those of other European countries, this is a major point of focus for educators and educationalists. In the German school system, whole-day education is not the rule in many federal states; hence, a broader pedagogical perspective, including both informal learning settings and social work, is not included on a regular basis, nor are approaches based on community education [84]. Hence, recent schooling policies in Germany do not respond appropriately to the needs of newly arrived adolescent immigrants. This can be interpreted as a structural problem embedded in overall educational and migration policies [85]. Considering migration as a potential 'master status' (depending on the individual status and the family situation) [86] for adolescent immigrants, this must become a major task for education and education policies in Germany. Otherwise, the school pathways for newly arrived immigrants will be limited to a low level of education and, hence, prosperity level [87].

**Funding:** Part of this research was funded by Max-Traeger-Stiftung, grant number MTS-Projekt: 5162-2014.

**Institutional Review Board Statement:** Ethic Committee Name: Ethik-Kommission der DGfE (Ethic committee of the German Educational Research Association (GERA)). Approval Code: 6 January 2022. Approval Date: 09/2021/DGFE.

**Informed Consent Statement:** Informed consent was obtained from all subjects involved in the study.

**Data Availability Statement:** The data presented in this study are available on request from the corresponding author. The data are not publicly available due to the vulnerable status of the participants as adolescents and refugees.

**Conflicts of Interest:** The author declares no conflict of interest.

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
