# Peer review of "Learning to Adapt? Leave and Arrival as Major Psycho-Social Challenges for Newly Arrived Adolescent Immigrants in Germany"

_2673-995X, doi:10.3390/youth3030052_

Round 1

Reviewer 1 Report

Comments

1.     The introduction has clearly outlined the background. There is one question in this part, as stated in the last paragraph, “The total number of interviews from both studies is 26, 15 interviewees had a so-called migration background, which means that at least one parent was not born in Germany, and 7 interviewees immigrated themselves with their families.” Please clarify the total number of the interview (e.g., 15 + 7 = 22, instead of 26? )

2.     The procedures of data analyses can be clearer and more detailed.

3.     The author outlined the difficulties and challenges of young immigrants in the Results. It would be good if the discussion and the implication could be in-depth.

4.     The limitation of this study should be discussed.

Author Response

Thank you very much for you review and helpful advise to the paper!

  • The introduction has clearly outlined the background. There is one question in this part, as stated in the last paragraph, “The total number of interviews from both studies is 26, 15 interviewees had a so-called migration background, which means that at least one parent was not born in Germany, and 7 interviewees immigrated themselves with their families.” Please clarify the total number of the interview (e.g., 15 + 7 = 22, instead of 26? )

Response: Sorry, that was unclear: I changed the paragraph:

"To reconstruct how learning is experienced by the young immigrants and at the same time how it is interwoven with socialisation, a secondary analysis of the samples of two qualitative interview studies was carried out. Both studies had an open qualitative design and included adolescents between 11 and 17 years. The first study focused on young adolescents with unspecific backgrounds concerning migration to reconstruct the interrelations of formal and informal learning [20,23], and the second study focused on newly arrived adolescents[24,25]. The total number of interviews from both studies is 26, 15 interviewees were immigrants, which means that at least one parent was not born in Germany, and 7 of these 15 interviewees immigrated themselves with their families. The interviews were analysed by using a reconstructive method [26,27]. Hence, all interviews included learning experiences, whereas most of the interviewees were positioned as migrants due to their families’ migration and 7 of them had personally experienced migration. In this paper, I will focus on the typical features of learning when migration and adolescence coincide." (73-86)

  1. The procedures of data analyses can be clearer and more detailed.

I added this paragraph to clearify:

"

The procedure of the analyses includes firstly the analysis of biographical data (dates, places, events) whithout the consideration of the subjective interpretations of the narrator. Secondly, the interviewtext is devided in sequences, according to the textstructure and -content. The aim of the analysis is to find out which mechanisms control the selection and design as well as the temporal and thematic linking of the text segments. The third step is the reconstruction of the case storyline according to the biographical data and the subjective way of telling the story. Forthly, particularly important interview sequences are subjected to a micro-analysis (meaning, word by word). Fifthly, the experienced biography and the told narration are contrasted (what is said in which way why?). After this a case structure hypothesis can be formulated. Finally, the analyses of several cases is condensed into a typology." (252-262)

3. The author outlined the difficulties and challenges of young immigrants in the Results. It would be good if the discussion and the implication could be in-depth.

I extended the discussion and the conclusion and added further references.

4. The limitation of this study should be discussed.

The limitations are included on page 6.

Reviewer 2 Report

Review

Thank you for the opportunity to review the paper entitled “Flight and Arrival as Major Psycho-Social Challenges for newly 3 arrived Adolescent Immigrants in Germany”.

This is overall a well-written article with an interesting focus. The strong point of this paper was the very clearly written German context. However, the paper in its current form is not yet ready for publication. Below I will explain my decision in further detail.

First, the title and abstract promise a study into refugee children and their experiences, coping strategies, and adolescent development. However, reading further, the sample consists of children of immigrants with unclear reasons for moving or non-refugee immigrants, and their coping strategies or development is hardly touched upon. The terminology of refugees and immigrants is quite unclear throughout the paper, while these are two different, and not always comparable, groups.

Second, I miss this paper's general research question and related focus. After reading the introduction, I miss the research question: what is it actually that you wish to study? Try to be clear about what this study examines and what it adds to the literature. From this question, trim the introduction down to the information that is relevant to understand the research problem, question, and relevance, and leave out all parts that are not relevant to this particular study.

In addition, learning is framed in the introduction as an all-encompassing learning experience, including both informal and formal learning processes. However, reading the results and the quotes of participants, all data seems to be focused on language skill development. What is the actual focus? These points add to the general feeling of a slight mismatch between the introduction and results, where the results do not follow the line of reasoning in the introduction.

Third, I have concerns about this study's research method and sample selection. First, the fact that this study used secondary data has its limitations, but the authors do not elaborate on how this could have affected their results. I miss information about the actual topic of these interviews. If these data were not collected for this specific topic, what then was the focus? Could this focus bias the results of this study? Second, in the introduction, the authors mention they use data from 24 interviews, collected in two studies, while in the results section, all results are based on a selection of only 7 of these interviews. This is a very limited number of participants to base your results on, so if the authors think that this number is sufficient, it should be explained why (for example, did you reach saturation with these limited numbers?) Please be transparent from the start about who your sample is, how many participants participated, what the selection criteria were, and how the sample size was decided. Third, although the methods are described in detail, I miss crucial information about why this method is chosen. What does this specific method add specifically to answer your research question? Fourth, there is no information provided about the studied participants: Who are they? How and where was this sample selected? Especially since the backgrounds of the participants, their migration trajectories, their country of origin, and their reasons for migration, are crucial for their learning experiences, an elaborated description of the sample is essential. Fifth, only adolescents who recently arrived in Germany are interviewed, but all interviews were held in German. How fluent were these adolescents in German after spending only 1-3 years in Germany?

Finally, the conclusion and discussion need to be more embedded in the literature and theory. Now, there is hardly any link with previous studies, theory, or other literature, which makes the conclusion too subjective for an empirical research article. Moreover, the conclusions drawn do not align with the results. For example, the role of languages seems crucial from the results, but is downplayed as ‘subordinate’ in lines 474-475, why?

Author Response

Thank you very much for the detailed and very helpful review! Here are my responses:

First, the title and abstract promise a study into refugee children and their experiences, coping strategies, and adolescent development. However, reading further, the sample consists of children of immigrants with unclear reasons for moving or non-refugee immigrants, and their coping strategies or development is hardly touched upon. The terminology of refugees and immigrants is quite unclear throughout the paper, while these are two different, and not always comparable, groups.

Response: Yes, the terminology was not coherent. Thank you for pointing at this. I went though the paper and amended the wording and the concepts in the whole paper. Since the I am focussing on newly arrived immigrants of different statuus. That some overreaching phenomena could be reconstructed is a crucial result of the considered data.

Second, I miss this paper's general research question and related focus. After reading the introduction, I miss the research question: what is it actually that you wish to study? Try to be clear about what this study examines and what it adds to the literature. From this question, trim the introduction down to the information that is relevant to understand the research problem, question, and relevance, and leave out all parts that are not relevant to this particular study.

Response: I added the research questions on page 2 (line 87-90)

In addition, learning is framed in the introduction as an all-encompassing learning experience, including both informal and formal learning processes. However, reading the results and the quotes of participants, all data seems to be focused on language skill development. What is the actual focus? These points add to the general feeling of a slight mismatch between the introduction and results, where the results do not follow the line of reasoning in the introduction.

Response: Of course, learning German is one of the major tasks for the newly arrived immigrants and it is strongly emphasised by the host society as I pointed out by referring to discrimination and the monolingual habitus of the German school. However, the data presented does not only and not primarily focus language acquisition as a single aspect; it is embedded since the self-representations are linguistic and related to power structures. Furthermore, I extended the discussion and the conclusion to explain the interrelations. I hope this relates better now.

Third, I have concerns about this study's research method and sample selection.

First, the fact that this study used secondary data has its limitations, but the authors do not elaborate on how this could have affected their results. I miss information about the actual topic of these interviews. If these data were not collected for this specific topic, what then was the focus? Could this focus bias the results of this study?

Response: I added information about the studies:

“All interviewees were enrolled in compulsory education. The contact was initialised via teachers, social workers or educators who enabled information events in school related but informal settings where I was able to explain my work and ask for participants. The interviewees later on contacted me to comply and take part. Time and space were determined by the participants. In the first study I asked all students to participate and in the first study only thos who were newly arrived in Germany. All were fluent in German. Those who were not might have decided not to participate even though translation was offered.

Even though the initial research questions of the studies were different and specifically aiming at learning, the data is appropriate to answer the research questions concerning learning experiences in the context of leave and arrival discussed here. Both studies were working with open interview designs, meaning open questions that invited the interviewees to speak about themselves and their stories. Moreover, the interviews were analysed reconstructively and hence word by word, closely to the text, so the inherent meaning was extracted and only in the next step contextualised with theory. Thus, that similar logics of experiences emerged across the studies, underlines the objectivity and reduces researcher’s bias (at least to a certain extent).” (208-224)

Second, in the introduction, the authors mention they use data from 24 interviews, collected in two studies, while in the results section, all results are based on a selection of only 7 of these interviews. This is a very limited number of participants to base your results on, so if the authors think that this number is sufficient, it should be explained why (for example, did you reach saturation with these limited numbers?) Please be transparent from the start about who your sample is, how many participants participated, what the selection criteria were, and how the sample size was decided.

Third, although the methods are described in detail, I miss crucial information about why this method is chosen. What does this specific method add specifically to answer your research question?

Response: I changed the section:

“To reconstruct how learning is experienced by the young immigrants and at the same time how it is interwoven with socialisation, a secondary analysis of the samples of two qualitative interview studies was carried out. Both studies had an open qualitative design and included adolescents between 11 and 17 years. The first study focused on young ado-lescents with unspecific backgrounds concerning migration to reconstruct the interrela-tions of formal and informal learning [20,23], and the second study focused on newly ar-rived adolescents[24,25]. The total number of interviews from both studies is 26, 15 inter-viewees were immigrants, which means that at least one parent was not born in Germany, and 7 of these 15 interviewees immigrated themselves with their families. The interviews were analysed by using a reconstructive method [26,27]. Hence, all interviews included learning experiences, whereas most of the interviewees were positioned as migrants due to their families’ migration and 7 of them had personally experienced migration. In this paper, I will focus on the typical features of learning when migration and adolescence coincide.” (73-86)

Fourth, there is no information provided about the studied participants: Who are they? How and where was this sample selected? Especially since the backgrounds of the participants, their migration trajectories, their country of origin, and their reasons for migration, are crucial for their learning experiences, an elaborated description of the sample is essential. Fifth, only adolescents who recently arrived in Germany are interviewed, but all interviews were held in German. How fluent were these adolescents in German after spending only 1-3 years in Germany?

 Response: I added:

“Even though the initial research questions of the studies were different and specifi-cally aiming at learning, the data is appropriate to answer the research questions con-cerning learning experiences in the context of leave and arrival discussed here. Both stud-ies were working with open interview designs, meaning open questions that invited the interviewees to speak about themselves and their stories. Moreover, the interviews were analysed reconstructively and hence word by word, closely to the text, so the inherent meaning was extracted and only in the next step contextualised with theory. Thus, that similar logics of experiences emerged across the studies, underlines the objectivity and reduces researcher’s bias (at least to a certain extent).” (216-224)

Finally, the conclusion and discussion need to be more embedded in the literature and theory. Now, there is hardly any link with previous studies, theory, or other literature, which makes the conclusion too subjective for an empirical research article. Moreover, the conclusions drawn do not align with the results.

For example, the role of languages seems crucial from the results, but is downplayed as ‘subordinate’ in lines 474-475, why?

Response: I extended the discussion and conclusion and changed the wording.

Reviewer 3 Report

Thank you very much for submitting your article. I really enjoyed reading it and think that it is well-done and explains interesting aspects. Especially the summarizing discussion is successful (however, the discussion part could be a little longer there could be a bit more about the power structures within the German school/educational system) 

With all the positive aspects, I miss one very important aspect. You interviewed minor refugees/immigrants: where is your positive ethics vote? before you interviewed the minor newcomers, you certainly submitted an ethics application  (especially with such a vulnerable group).

Other points:  

- line 24/23: please specify, where did they come from? why? and very important:  whether and how many of them were unaccompanied minor refugees ? (in the course of the article this information may be relevant)

- line 43: This sentence seems as if on the integration task is a one-way street. by whom was the 'integration task' missed?

-line 75/56: migration-background: please discuss the term critically, recent recommendations of the "Fachkommission Integrationsfähigkeit"strongly advocate not to use the term anymore, instead other terms are suggested. please revise. 

- general recommendation: please edit the formal aspects and grammar, for example in line 80 there is one dot too many. 

Thanks! 

Author Response

Thank you very much for your helpful and important remarks!

Here are my responses:

Especially the summarizing discussion is successful (however, the discussion part could be a little longer:  there could be a bit more about the power structures within the German school/educational system)

Response: I extended the discussion and conclusion

With all the positive aspects, I miss one very important aspect. You interviewed minor refugees/immigrants: where is your positive ethics vote? before you interviewed the minor newcomers, you certainly submitted an ethics application (especially with such a vulnerable group).

Response: Yes, I did.

Other points: 

- line 24/23: please specify, where did they come from? why? and very important:  whether and how many of them were unaccompanied minor refugees ? (in the course of the article this information may be relevant)

Response: There are no unaccompanied minors in my samples. I made this more explicit.

- line 43: This sentence seems as if on the integration task is a one-way street. by whom was the 'integration task' missed?

Response: I explained this:

“The German education system is highly selective since pupils are separated into different tracks already after grade 4 [10,11]. The concrete configuration differs between the 16 federal states, yet, in most of them exist two to three different tracks and only one leads to further academic education. For newly arrived immigrants so-called Welcome or Preparation Classes were established since 2015, mainly to provide German language skills [12]. These classes are mainly situated at primary and lower-track schools [13]. Recent figures suggest that young immigrants are underrepresented in academic tracks and higher education (ibid.). Hence, the goal to integrate immigrants in terms of equal achievement and educational participation was missed.

This situation can be tackled successfully only if we know what must be done in or-der to achieve integration, not only in terms of the immigrant’s formal education but also in terms of their informal educational experiences. For adolescents who leave their countries, the context is one in which formal education has been interrupted or even stopped altogether [14–19]. Given this, researchers need to consider the learning experiences of adolescent immigrants from a broader perspective beyond that of formal education, in the context of social conditions to which they are exposed. This perspective follows current educational research on the importance of recognising both informal and formal learning among adolescent immigrants [20].”

-line 75/56: migration-background: please discuss the term critically, recent recommendations of the "Fachkommission Integrationsfähigkeit"strongly advocate not to use the term anymore, instead other terms are suggested. please revise.

Response: Thank you for this information! I revised it.

- general recommendation: please edit the formal aspects and grammar, for example in line 80 there is one dot too many.

Response: I went through the paper and made corrections.

Reviewer 4 Report

"Learning to Adapt" is a thoughtful contribution to the literature on adolescent migration focused on the German reception context. I have a few suggestions for the author to consider in order to improve the manuscript

Methods:

The methods are well described, particularly around the use of Rosenthal’s method. I would like to know more about how the cases discussed reflect the broader pool of participants. In particular, what were the participants’ ages and the pool’s age ranges at the time of the interview (and, if applicable at the time of migration); what was the breakdown of national origins, racial backgrounds, and gender of pool. This information would situate the cases in the broader context of the research and provide stronger evidence for the claims made.

The author argues for a kind of unified experience of migration, while also situating this claim in an understanding that social inequalities matter. This participant/pool information would bolster this claim that the experience of migration is a prime determinate of learning given that across observed differences of identity and background migration nevertheless emerges as this prime determinate of learning, with added difficulties given the broader contexts of that migration (starker for those fleeing war and the risk of death versus dire economic straights). I would also suggest grounding this claim by discussing Roberto Gonzales’ work on immigration status as a sociological “master status” for life outcomes. It seems to me the author is making a similar claim with respect to being an immigrant youth regardless of status.

Findings:

The findings regarding the flipping of generational responsibilities because of German language acquisition could also be better grounded in the literature and expanded upon a bit more. Marjorie Orellana’s work in the US context Inmaculada Garcia-Sanchez’s work in Spain is essential here.

Discussion:

The author discusses the migration decisions made in terms of participants’ mothers’ choices in stark language (lines 436-439). I’m wondering if the author could speak to why they highlighted the role of mothers here? It seems that in the cases discussed the mother made the migration decision. Was that broadly the case across the broader pool? Or, does the fact that for these focal cases mothers making the migration decision mattered in a  strong way that informed the learning experience of these specific youth? Given the centrality of intergenerational relations to the framing, and this strong positioning of mothers, I think it is worth differentiating these relations a bit more to see if there are differences according to gender. Rhacel Parrenas’s work and that of Pierrette Hondagneu-Sotelo with respect to Fillipina care workers could be informative.

Minor Points:

On line 43, the author states that the goal to integrate immigrants was missed in the tracking of immigrant youth in German schools. I would restate. The goal of integrating them for chances of socio-economic mobility was lost. They were integrated, just into an underclass. See Jeannie Oakes work on tracking in US schools.

In the theoretical framework, you introduce King’s work twice. This could be consolidated.  

Author Response

Thank you very much for your helpful comments! I particularly appreciate the literature you recommend since it broadened and deepened not only the paper but my overall research-perspective.

Here are my responses to your comments:

Methods:

 The methods are well described, particularly around the use of Rosenthal’s method. I would like to know more about how the cases discussed reflect the broader pool of participants. In particular, what were the participants’ ages and the pool’s age ranges at the time of the interview (and, if applicable at the time of migration); what was the breakdown of national origins, racial backgrounds, and gender of pool. This information would situate the cases in the broader context of the research and provide stronger evidence for the claims made.

Response: I gave more detailed information about my interviewees:

“The two studies this paper builds on were both not originally focusing on the learning experiences of newly arrived adolescents. The first one aimed to explore the interrelations of formal and informal learning in early adolescence [20] and the second one was a pilot study on adolescent immigrants and their experiences in the context of migration and arrival in Germany [24,65,77]. Whereas the first study included immigrants and non-immigrants, the second focused on newly arrived young people. The national back-grounds of the adolescents were diverse: some came from Eastern Europe, some from the Middle East, and some from Africa. Both studies used the same methodical design, namely narrative interviews [78] that encouraged the adolescents to tell their stories. All interviews were conducted in German. It was explained that the participation was voluntary and can be cancelled at any time. The young people chose the place and time of the interview and if they wanted to be accompanied by a friend or family member.

All interviewees were enrolled in compulsory education. The contact was initialised via teachers, social workers or educators who enabled information events in school related but informal settings where I was able to explain my work and ask for participants. The interviewees later on contacted me to comply and take part. Time and space were deter-mined by the participants. In the first study I asked all students to participate and in the first study only those who were newly arrived in Germany. All were fluent in German. Those who were not might have decided not to participate even though translation was offered.” (197-215)

The author argues for a kind of unified experience of migration, while also situating this claim in an understanding that social inequalities matter. This participant/pool information would bolster this claim that the experience of migration is a prime determinate of learning given that across observed differences of identity and background migration nevertheless emerges as this prime determinate of learning, with added difficulties given the broader contexts of that migration (starker for those fleeing war and the risk of death versus dire economic straights). I would also suggest grounding this claim by discussing Roberto Gonzales’ work on immigration status as a sociological “master status” for life outcomes. It seems to me the author is making a similar claim with respect to being an immigrant youth regardless of status.

Response: I included this work in my conclusion.

Findings:

The findings regarding the flipping of generational responsibilities because of German language acquisition could also be better grounded in the literature and expanded upon a bit more. Marjorie Orellana’s work in the US context Inmaculada Garcia-Sanchez’s work in Spain is essential here.

Response: I included the work and tried to emphasis the interrelation of language acquisition and learning.

Discussion:

The author discusses the migration decisions made in terms of participants’ mothers’ choices in stark language (lines 436-439). I’m wondering if the author could speak to why they highlighted the role of mothers here? It seems that in the cases discussed the mother made the migration decision. Was that broadly the case across the broader pool? Or, does the fact that for these focal cases mothers making the migration decision mattered in a  strong way that informed the learning experience of these specific youth? Given the centrality of intergenerational relations to the framing, and this strong positioning of mothers, I think it is worth differentiating these relations a bit more to see if there are differences according to gender. Rhacel Parrenas’s work and that of Pierrette Hondagneu-Sotelo with respect to Fillipina care workers could be informative.

Response: Thank you very much for making this point! I referred to the family dynamics know and hope this clearer now. However, I decided not to discuss gender here and stay on a more general level of parenthood since I think this would be a new aspect. This very important and needs further research from my side on the German data.

Minor Points:

On line 43, the author states that the goal to integrate immigrants was missed in the tracking of immigrant youth in German schools. I would restate. The goal of integrating them for chances of socio-economic mobility was lost. They were integrated, just into an underclass. See Jeannie Oakes work on tracking in US schools.

Response: Yes, that is true! We observe similar developments. I added this at the end of the paper.

In the theoretical framework, you introduce King’s work twice. This could be consolidated.

Response: I took this section out.

Round 2

Reviewer 2 Report

Thank you for the opportunity to re-review the paper entitled “Flight and Arrival as Major Psycho-Social Challenges for newly arrived Adolescent Immigrants in Germany”. 

The authors received reviewers comments and revised the manuscript accordingly. Initially, I had my concerns about the design of the study, which is something that the authors, understandably, could not adjust in this short time frame. Although the authors adjusted the manuscript according to reviewers comments, in my opinion, this so far did not improve the manuscript to the high standards of scientific publications.

First, my most important concern is related to the study design and the lack of transparency about the research data. That is, the limitation of secondary data has not been satisfactorily addressed in the current manuscript. This means that I still have my concerns about the selectivity of the data: Since both studies were not about the study topic, the data is likely to lack depth and richness. The authors now state that data are appropriate to study their research question, but not why that would be the case. Moreover, the sample description was lacking in the first manuscript version, but the revisions did not add all necessary information (e.g., "some came from Eastern Europe, some from the Middle East, and some from Africa" --> how many, where in Eastern Europe?). In addition, it is still not clear to me how many interviews are analyzed for this particular study. In the introduction, it is mentioned that 26 interviews were used, while in the results only 7 cases seem to appear. 

Second, the manuscript still lacks clear language throughout (e.g., some sentences are so long, full of complicated jargon, and not-generally known constructs that it is difficult to follow, even for other social scientists). Some revisions seem to have been written hastily, thereby not improving the overall language quality of the manuscript. For example, now the word 'flight' has been changed into 'leave' in some parts of the text, but not in the title. Overall language is not scientific but often turns into subjective and implicit language (e.g., the use of 'very' twice in one sentence, while the word 'very' is vague and not objective). 

I hope the authors will continue with this interesting topic, but search for further improvement of the current manuscript. Collecting additional data would improve the foundation of the study and its contribution. 

Author Response

Thank you for your secend review. First of all, yes it was a short time and I admit that due to my workload I might hav been too hasty. I know tried to be more senisitive accoring to language and style.

Coming to your amendments:

"First, my most important concern is related to the study design and the lack of transparency about the research data. That is, the limitation of secondary data has not been satisfactorily addressed in the current manuscript. This means that I still have my concerns about the selectivity of the data: Since both studies were not about the study topic, the data is likely to lack depth and richness. The authors now state that data are appropriate to study their research question, but not why that would be the case. Moreover, the sample description was lacking in the first manuscript version, but the revisions did not add all necessary information (e.g., "some came from Eastern Europe, some from the Middle East, and some from Africa" --> how many, where in Eastern Europe?)."

I added the concrete number. Yet, I do not mean to be intransparent. but my focus is not the national background, but the status and situation in the country and learning settings of arrival. The presented results emerged from a reconstructive analysis which is explained in the paper. Resulting from this, the data may be reused in particular because the research questions and the interview questions are open. If this would not by possible, new and unexpected results would never appear. This does not relate to a positivistic approach, but surely to Pierce's idea of upduction. Hence, I would insist, that my reseaerch procedure is both legitimate and transparent.

"In addition, it is still not clear to me how many interviews are analyzed for this particular study. In the introduction, it is mentioned that 26 interviews were used, while in the results only 7 cases seem to appear."

All 26 interview were analysed and the results are already published in peer reviewed journals. The 7 interviews discussed in this paper were chosen, because all of them are newly arrived immigrants and the reconstructive analyses showed structural ovrreaching aspects that I present in this paper.

"Second, the manuscript still lacks clear language throughout (e.g., some sentences are so long, full of complicated jargon, and not-generally known constructs that it is difficult to follow, even for other social scientists). Some revisions seem to have been written hastily, thereby not improving the overall language quality of the manuscript. For example, now the word 'flight' has been changed into 'leave' in some parts of the text, but not in the title. Overall language is not scientific but often turns into subjective and implicit language (e.g., the use of 'very' twice in one sentence, while the word 'very' is vague and not objective). "

Sorry, that you have this impression. I will reread it and try to improve it.

Reviewer 3 Report

Thank you for your detailed answers and further explanations which are very helpful. 

"With all the positive aspects, I miss one very important aspect. You interviewed minor refugees/immigrants: where is your positive ethics vote? before you interviewed the minor newcomers, you certainly submitted an ethics application (especially with such a vulnerable group)."

"Response: Yes, I did."

You answer this question with yes, but in your manuscript a statement with a positive ethics vote as well as with the responsible ethics committee is missing. Please add this important point. 

For example in this form: 

Ethical approval: The study was approved by the Ethics Commission of ... (Date: XX  and Number: XX).

Author Response

Thank you so much! I am sorry: I added the ethics approval to the editors, but not in the manuscrpit you rivised. It is now in the paper.

Reviewer 4 Report

I appreciate how well the author integrated feedback, particularly the new literature. A few points remain before publication:

Introduction

I appreciate the breakdown of the broader study. A clear line here that the author is looking specifically at the 7 immigrant youth in particular (and not the second generation youth) would be helpful.

Literature Review

I perhaps would not leave Gonzales to the end, but integrate here. Also, he is talking about immigration status (authorized v. unauthorized) as a master status. I think the author is extending this argument to say being an immigrant generally is a master status—regardless of legal status held (e.g., refugee v. economic migrant, authorized v. unauthorized). I would add that to the literature review to bolster the claim that adolescent migration is a definitional experience for development. I would also then reword the final reference to Gonzales to reflect that intervention. Also, that line is a bit difficult to parse as currently written.

Methods

I would still like to know the ages and the pool’s age ranges at the time of the interview (and, if applicable at the time of migration) of the focal youth and the gender breakdown. The new additions on lines 209-2015 are excellent to be included; however, line 212-213 appears in the above paragraph. I would suggest the author go through and make sure all the additions are aligned with the prior text.

Text edits

Line 25: should be unaccompanied

Line 26: Should be them

Line 30: should be immigrants

Lines 112 and 127: Should this be refugee or immigrant given the shifted focus of the paper?

Line 495: Should be institutions

Line 543: Should be master

Author Response

Thank for your concrete amendments! I will resond to them diectly.

Introduction

I appreciate the breakdown of the broader study. A clear line here that the author is looking specifically at the 7 immigrant youth in particular (and not the second generation youth) would be helpful.

=> I added this in line 92.

Literature Review

I perhaps would not leave Gonzales to the end, but integrate here. Also, he is talking about immigration status (authorized v. unauthorized) as a master status. I think the author is extending this argument to say being an immigrant generally is a master status—regardless of legal status held (e.g., refugee v. economic migrant, authorized v. unauthorized). I would add that to the literature review to bolster the claim that adolescent migration is a definitional experience for development. I would also then reword the final reference to Gonzales to reflect that intervention. Also, that line is a bit difficult to parse as currently written.

=> I added the reference in the theory chapter in lines 107/108; 131/132; 205-209 and I chanced the sentence in the end (line 578 f).

Methods

I would still like to know the ages and the pool’s age ranges at the time of the interview (and, if applicable at the time of migration) of the focal youth and the gender breakdown. The new additions on lines 209-2015 are excellent to be included; however, line 212-213 appears in the above paragraph. I would suggest the author go through and make sure all the additions are aligned with the prior text.

=> I changed that section, now lines 232 ff. I also added the age range.

Text edits

Line 25: should be unaccompanied

Line 26: Should be them

Line 30: should be immigrants

Lines 112 and 127: Should this be refugee or immigrant given the shifted focus of the paper?

Line 495: Should be institutions

Line 543: Should be master

Thanky you, I corrected all. Best regards!